# Effect of *Cudrania tricuspidata* on Cariogenic Properties and Caries-Related Gene Expression in *Streptococcus mutans*

**DOI:** 10.3390/molecules30081755

**Published:** 2025-04-14

**Authors:** Eun-Sook Kim, Ji-Eon Jeong, Young-Hoi Kim, Yong-Ouk You

**Affiliations:** 1Institute of Biomaterials and Implant, College of Dentistry, Wonkwang University, Iksan 54538, Republic of Korea; bacteria@wku.ac.kr; 2Department of Oral Biochemistry, School of Dentistry, Wonkwang University, Iksan 54538, Republic of Korea; jje444@naver.com; 3Department of Food Science and Biotechnology, Jeonbuk National University, Jeonju 54896, Republic of Korea; yhoil307@hanmail.net; 4Wonkwang Dental Research Institute, Wonkwang University, Iksan 54538, Republic of Korea

**Keywords:** *Cudrania tricuspidata*, *Streptococcus mutans*, biofilm, virulence genes, UPLC-Q-TOF-MS

## Abstract

The purpose of this study was to evaluate the gene expression pattern of the caries-inhibiting effect of *Cudrania tricuspidata* (*C. tricuspidata*) extract on cariogenic bacteria *Streptococcus mutans* (*S. mutans*). We examined bacterial growth, tooth surface attachment, biofilm formation, acid production, free calcium release, and toxicity gene expression. The major components of the extract were investigated by UPLC-Q-TOF-MS analysis. Exposure to *C. tricuspidata* inhibited bacterial growth and attachment at concentrations of ≥15 μg/mL. Inhibition effects on biofilm formation, acid production, and free calcium release due to acid production were observed at concentrations ≥ 30 μg/mL. *S. mutans* virulence gene expression analysis showed that it inhibited the expression of *gbpB* and *spaP*, which mediate bacterial attachment to the tooth surface, and that of genes contributing to biofilm formation (*gtfB*, *gtfC*, and *gtfD*) and acid resistance (*brpA* and *relA*), and regulation (*vicR*). Analysis using UPLC–Q–TOF–MS/MS showed that the main component was phenylpropanoids. These results suggest that *C. tricuspidata* may inhibit the cariogenic properties associated with the expression of caries-related genes in *S. mutans*.

## 1. Introduction

Dental biofilm formation is important in the development of dental caries. It is formed through a multi-step process. The first step involves the formation of an acquired pellicle on the tooth surfaces. Subsequently, primary colonizers, such as Gram-positive streptococci (e.g., *Streptococcus mutans* [*S. mutans*], *Streptococcus oralis*, *Streptococcus sanguinis*, and *Streptococcus mitis*), as well as Neisseria species, adhere to the surface of the acquired pellicle. Dental biofilms progressively form through the proliferation of primary colonizing bacteria and the aggregation and adhesion of secondary colonizing microorganisms. Secondary colonizers mainly consist of Gram-negative bacteria such as *Fusobacterium nucleatum*, *Prevotella intermedia*, Actinomyces species, and Capnocytophaga species [1]. Among the microorganisms present in the dental biofilms, mutans streptococci related to dental caries include *S. mutans*. Once sucrose has been broken down into glucose and fructose, *S. mutans* releases glucosyltransferase (GTF), which allows glucose to form dental plaques. These plaques calcify and serve as a matrix for the fixation of the bacterium, while fructose induces demineralization in the teeth, which become susceptible to dental caries, as pH is lowered by lactic acids produced by *S. mutans* [2,3,4]. Virulence factors of *S. mutans*, which include bacterial adherence, extracellular polysaccharide formation, biofilm formation, glucose absorption and metabolism, acid tolerance, and regulation and involvement, contribute to disease development [5]. Genes related to adherence include *spaP*, which interacts with salivary agglutinin glycoprotein, and *gbpB*, which affects the tooth surface adherence of glucan, both of which affect *S. mutans* adherence to the tooth surface [6,7]. Glucan synthases GTFB, GTFC, and GTFD are produced by *gtfB*, *gtfC*, and *gtfD* and regulated by *vicR* [8,9]. In addition, *brpA* and *relA* stabilize biofilm formation and are involved in the glucose phosphotransferase system, which mediates acid tolerance [10,11,12]. Overall, this evidence suggests that suppressing the growth of *S. mutans* and associated virulence factors may help prevent the onset of dental caries.

Previous studies have shown that lotus leaf, dandelion, and prickly pear extracts inhibited the growth of *S. mutans* [13,14]. Some studies have reported the inhibition of *S. mutans* growth, acid production, adherence, and insoluble glucan synthesis, as mediated by *Caesalpinia sappan* and *Aconitum koreanum* extracts [15,16]. In addition, *Diospyros malabarica* stem extracts had an inhibitory effect on biofilm formation associated with *S. mutans* [17].

The *Cudrania tricuspidata* (*C. tricuspidata*) is a plant mainly found in East Asia, including Korea, China, and Japan, and has long been used as a folk remedy. In traditional Korean medicine, it has been used as a nutritional tonic, helping to improve insomnia, hangover, vision, and liver and kidney function. The Dictionary of Traditional Chinese Medicine describes it as effective in treating joint pain, jaundice, and swelling [18,19]. The physiological activity of *C. tricuspidata* is mediated by xanthones and flavonoids. Isolated prenylated xanthones show anti-obesity and anti-diabetes effects based on protein tyrosine phosphatase 1B (PTP1B) inhibition [20]. Cudratricusxanthone A demonstrates strong antiproliferative, antioxidant, and monoamine oxidase inhibitory effects [21]. The prenylated isoflavone skeleton has been reported to be key to antibacterial activity [22]. However, their inhibitory effect on dental caries has not been examined previously. Hence, this study examined the inhibitory effect of *C. tricuspidata* extracts on *S. mutans* growth, adherence, biofilm formation, organic acid production, free calcium release, and virulence gene expression and subsequently evaluated their potential to inhibit dental caries.

## 2. Results

### 2.1. UPLC-Q-TOF-MS Analysis Results of C. tricuspidata Extract

As a result of analyzing the ethanol extract of *C. tricuspidata* roots by UPLC Q-TOP, 48 major components were identified (Table 1). The isomer components analyzed were isoarundinin II, cudratricusxanthone L, gericudranin E, kuwanon E, cudraflavenone A, cudraxanthone B, mortatarin B, and mortatarin C (Table 1). The total ion mass spectrum of *C. tricuspidata* extract in negative ion mode is shown in Figure 1.

### 2.2. Inhibitory Effect on S. mutans Growth

The inhibitory effect of ethanol extracts of *C. tricuspidata* on *S. mutans* growth was examined at concentrations of 15, 30, 45, and 60 μg/mL, which showed 13%, 50%, 87%, and 96% inhibition, respectively, compared to the growth rate of 100% in the control group that was not treated with the extracts. Chlorhexidine (CHX), which was used as the positive control, showed 99% inhibition of growth at a concentration of 0.05% (Figure 2).

### 2.3. Inhibitory Effect on S. mutans Biofilm Formation

The inhibitory effect of *C. tricuspidata* extracts on *S. mutans* biofilm formation was examined. The dish and artificial teeth observed under the scanning electron microscope showed that higher concentrations of the extracts resulted in a higher inhibition rate of *S. mutans* biofilm formation compared to that observed in the negative control group (Figure 3, Figure 4 and Figure 5). In particular, the inhibitory effect of the extracts on *S. mutans* biofilm formation was the highest at concentrations ≥45 μg/mL (Figure 3). The effect of the extracts on the expression of *gtfB*, *gtfC*, and *gtfD*, which affect *S. mutans* biofilm formation, and *vicR*, which regulates these genes, was measured via real-time PCR. The expression of *gtfB* was inhibited at ≥60 μg/mL. The expression of *gtfC* and *gtfD* was inhibited by the extract in a concentration-dependent manner at ≥15 μg/mL and ≥30 μg/mL, respectively, while the extract concentrations ≥30 μg/mL inhibited the expression of *vicR* (Figure 6).

### 2.4. Inhibitory Effect on S. mutans Adherence

We analyzed the inhibitory effect of the extracts on saliva-coated hydroxyapatite (S-HA) bead adherence measurement. *C. tricuspidata* extracts at concentrations of 15, 30, 45, and 60 μg/mL showed 40%, 60%, 82%, and 97% inhibition of adherence, respectively, compared to 0% inhibition observed in the negative control group (Figure 7). The expression of *gbpB* and *spaP* involved in mediating *S. mutans* adherence was observed in real-time PCR. The expression of *gbpB* and *spaP* was inhibited at concentrations ≥15 μg/mL and ≥30 μg/mL, respectively (Figure 8).

### 2.5. Inhibitory Effect on Organic Acid and Free Calcium Production by S. mutans

*S. mutans* were inoculated with different concentrations of the extracts to analyze the inhibitory effect of C. tricuspidata extracts on organic acid production. The pH was measured before incubation and after 24 h of incubation. A pH of 7.33 ± 0.04 in the negative control group and that of 5.39 ± 0.10 was observed after 24 h of incubation. At a concentration of 15 μg/mL, the extract showed a pH of 5.32 ± 0.012, which was comparable to that in the control group. The concentrations of 30 μg/mL, 45 μg/mL, and 60 μg/mL yielded pH values of 6.26 ± 0.09, 7.13 ± 0.04, and 7.17 ± 0.07, respectively. At concentrations ≥30 μg/mL, the extract inhibited organic acid production (Table 2).

We measured free calcium levels after a reaction in which hydroxyapatite/tricalcium phosphate (HA/TCP) was dissolved by organic acid released by *S. mutans*. We observed 24.6 μM of free calcium in the negative control group. At each concentration, the extract inhibited free calcium release (Figure 9). The expression levels of *relA* and *brpA*, which contribute to *S. mutans* acid tolerance, were examined by real-time PCR. The results showed that the expression of *relA* and *brpA* was inhibited at concentrations ≥15 μg/mL (Figure 10).

## 3. Discussion

Dental caries are common and associated with the interaction of bacteria in the dental plaque, as well as food and saliva components in the mouth. *S. mutans* is a major bacterium linked to the onset of dental caries. Treatments with CHX and fluoride can be used to prevent dental caries; however, they may lead to tooth discoloration, dental calculus formation, oral mucosal peeling, and the development of resistant bacterial strains. Remedies based on natural substances have been proposed to address these limitations [23].

The major components of the ethanol extract of *C. tricuspidata* analyzed by UPLC–Q–TOF–MS/MS were phenylpropanoids (14.19%), flavonoids (9.83%), and xanthones (6.54%). *C. tricuspidata* leaf and root bark extracts, obtained with methanol using a disk diffusion test, showed an antibacterial effect against *S. mutans* [24], and the component analysis results were similar to our analysis [20]. Isoarundinin II, which was first isolated from *Arundina bambusifolia*, belongs to phenylpropanoids and has been reported to exhibit anti-inflammatory, anticancer, and other inhibitory activities, including the inhibitory effect of NO production [25] and potent DPPH free radical scavenging effect [26]. Cudraatricusxanthone L has been reported to have antineuro–inflammatory effects [27]. The anticancer and antioxidant effects of flavanone gericudranin E [28,29], the antibacterial effect of kuwanone E [30], and the inhibitory effect of cudraflavenone A on pathogenic bacteria and biofilm formation [31] have been reported.

This study analyzed the inhibitory effect of *C. tricuspidata* extracts on dental caries induced by *S. mutans*. We observed the effects of the extract on bacterial growth, adherence, organic acid production, free calcium release, and biofilm formation and confirmed the expression patterns of virulence genes related to the development of dental caries by *S. mutans* via real-time PCR. At concentrations of 15, 30, 45, and 60 μg/mL, *C. tricuspidata* extracts showed 13%, 50%, 87%, and 96% inhibition of bacterial growth, respectively, compared to that observed in the control group. Compared to the growth inhibition of 31%, 95%, 98%, and 99.6% observed at concentrations of 125, 250, 500, and 1000 μg/mL of methanol extracts of *Caesalpinia sappan*, respectively [16], the inhibitory effect of *C. tricuspidata* extracts was more potent. For tooth surface adherence, *S. mutans* releases three types of GTF enzymes. GTF-S produces soluble glucan, while GTF-I and GTF-SI produce insoluble glucan. Three types of *gtf* genes are involved in the production of these enzymes: *gtfB*, *gtfC*, and *gtfD* produce GTF-I, GTF-SI, and GTF-S, respectively. Glucan is an insoluble polysaccharide synthesized by the aforementioned enzymes. Insoluble glucan turns into an underlying substance for the formation of dental plaque and helps *S. mutans* to adhere to the tooth surface [9,32]. *C. tricuspidata* extracts inhibited *S. mutans* tooth surface adherence (up to 40–97%) in a concentration-dependent manner. These effects were more potent than those of 16, 31, 63, and 125 μg/mL of *Aconitum koreanum* ethanol extracts with adherence inhibition of 5%, 26%, 50%, and 54%, respectively [15]. The key proteins of the surface protein antigen P (*spaP*), glucosyltransferase B (*gtfB*), and glucan-binding protein B (*gbpB*) are produced by *S. mutans* and are essential mechanisms for biofilm formation. *spaP* interacts with salivary α-galactosides and plays a key role in the early stages of biofilm development [33]. However, the expression of *C. tricuspidata* extracts was suppressed at a lower concentration (≥45 μg/mL) than other genes (*gtfC*, *gtfD*, *gbpB*, *brpA,* and *relA)*. The reason is presumed to be due to the effect of planktonic culture conditions, not the presence of saliva in the oral cavity. After the initial biofilm action of *spaP*, *S mutans* adhere to the tooth surface via the *gtfB* gene, facilitating the synthesis of extracellular glucans. This binding plays a key role in plaque formation. Refs. [9,34] Unlike *gtfB*, *gbpB* binds to existing glucans formed by *gtfB*, enhancing the initial binding of *S mutans* to the tooth surface. Therefore, key genes such as *gtfB*, *gbpB*, and *spaP* can be considered indicators for the development of new drugs aimed at treating dental caries [33]. The low expression of *spaP* and *gtfB* compared to other genes (*gtfC*, *gtfD*, *gbpB*) by Cudrania japonica is thought to be due to the influence of planktonic culture. However, the excellent inhibition of biofilm formation by *C. tricuspidata* is thought to be due to the interaction of *gtfC*, *gtfD*, and *gbpB* genes. Furthermore, *S. mutans* metabolizes glucose, produces organic acids, decreases pH in the oral cavity owing to acid production, and shows biofilm formation [35,36]. *C. tricuspidata* extracts at concentrations of ≥30 μg/mL infused above the dental caries threshold of pH 5.5 inhibited organic acid production by *S. mutans*. The expression of *brpA* and *relA* involved in acid tolerance declined in a concentration-dependent manner. Furthermore, with decreased organic acid production, the inhibitory effect of the extracts on free calcium released by HA/TCP likely contributed to the reduction in the occurrence of dental caries. The inhibitory effects of *C. tricuspidata* extracts on *S. mutans* growth, adherence, biofilm, and acid production were stronger than those of propolis extracts, which are considered beneficial for oral health [37]. In addition, while propolis extracts did not inhibit the expression of *gtfB*, *gtfC*, and *gtfD* involved in biofilm formation, *C. tricuspidata* extracts inhibited the expression of these genes, thereby decreasing dental plaque formation, and helping reduce the risk of dental caries [37].

Overall, *C. tricuspidata* extracts showed an inhibitory effect on *S. mutans* growth, adherence, organic acid production, free calcium release, biofilm formation, and virulence gene expression and are expected to help inhibit the onset of dental caries. These effects may be partly attributed to the major components of *C. tricuspidata*, namely phenylpropanoids, xanthones, and flavonoids, but further analysis of the components is required.

## 4. Materials and Methods

### 4.1. Material Extraction

*C. tricuspidata* roots were provided by the Jinan Kujibong Farm and identified by Dr. Seung Il Jeong at the Jeonju Agrobio-materials Institute (Jeonju, Republic of Korea). A reference specimen (no: 2-06-22) was stored in the Herbarium of the Department of Oral Biochemistry, School of Dentistry, at Wonkwang University. A total of 1500 mL of 95% ethanol was added to 100 g of *C. tricuspidata*, which was then extracted at room temperature for 72 h. The extracts were filtered, concentrated, dried, and refrigerated (−70 °C). They were thawed for every experiment and diluted with dimethyl sulfoxide to obtain different concentrations before use.

### 4.2. Bacterial Culture

The strain used in this experiment was inoculated with *S. mutans* ATCC 25175 at a concentration of 1 × 10^8^ CFU/mL in brain heart infusion (BHI, Difco, Sparks, MD, USA) broth, and after adding *C. tricuspidata* extracts by concentration (15, 30, 45, and 60 μg/mL), was then incubated at 37 °C for 24 h. Then, the optical density was measured with a 550 nm spectrophotometer. Moreover, 0.05% chlorhexidine (CHX) was used as a positive control. The experiment was repeated three times.

### 4.3. Biofilm Formation Assay

Different concentrations (15, 30, 45, and 60 μg/mL) of *C. tricuspidata* extracts were added to the BHI liquid medium containing 1% sucrose, and the medium was inoculated with 5 × 10^5^ CFU/mL of *S. mutans*. Artificial teeth (Endura, Shofu Inc., Kyoto, Japan) were placed in a 24-well plate, followed by incubation at 37 °C for 24 h. In addition, 35 mm dish cultures were cultured under the same strains and culture conditions as the artificial teeth. After incubation, the medium was removed, and the artificial teeth and dishes were washed with purified water, stained with 1% safranin for 30 s, washed again, and dried. The bound safranin at the bottom of dishes was released by 30% acetic acid, and the optical density of the solution was determined at 530 nm. The artificial teeth were photographed. The extracts were not added to the control group [38,39].

### 4.4. Scanning Electron Microscopic Analysis of S. mutans Biofilm Formation

The BHI liquid medium containing 1% sucrose was dispensed into a 35 mm dish. *C. tricuspidata* extracts (15, 30, 45, and 60 μg/mL) were added, and the bacterial solution was inoculated at the concentration of 5 × 10^5^ CFU/mL and incubated at 37 °C for 24 h. The incubated solution was removed, followed by cleaning with purified water. The solution was fixed after reaction with 2.5% glutaraldehyde (0.1 M sodium cacodylate buffer, pH 7.2) at 4 °C for 24 h. It was dehydrated with 70%, 80%, 95%, and 100% of ethanol, lyophilized, coated with gold, and imaged using the scanning electron microscope (SEM, Hitachi S-4800, Tokyo, Japan).

### 4.5. Bacterial Adherence

To analyze the effect of *C. tricuspidata* on bacterial adherence, we used the method reported by Liljemark et al. [40]. Hydroxyapatite beads (diameter 80 μm, Bio-Rad, Hercules, CA, USA) were coated with purified human saliva. The beads (S-HA) were then mixed with various concentrations (15, 30, 45, and 60 μg/mL) of *C. tricuspidata* in the bacterial suspension (1 × 10^7^ CFU/mL). After mixing gently, incubation was performed at 37 °C for 90 min. Then, the process of centrifuging at 10,000 rpm for 1 min and washing the S-HA with 1× PBS was repeated twice to remove non-attached bacteria and transferred to a new tube containing potassium phosphate buffer. *S. mutans* adhered to S-HA was dispersed at 50 W for 30 s using a sonicator (Fisher Scientific, Springfield, NJ, USA), and the supernatant was inoculated into the MSA plate medium (supplemented with 3.2 mg/mL of bacitracin). Colony-forming units were calculated after 48 h of incubation

### 4.6. Analysis of Organic Acid Production and Calcium Release by S. mutans

#### 4.6.1. Analysis of Organic Acid Production

The effect of *C. tricuspidata* (15, 30, 45, and 60 μg/mL) on acid production by *S. mutans* was evaluated as previously reported [13]. Acid production was measured in the medium before and after bacterial incubation using a pH meter (METTLER TOLEDO 320 pH meter, Shanghai, China). The extracts were not added to the control group.

#### 4.6.2. Analysis of Calcium Release

After *C. tricuspidata* (15, 30, 45, and 60 μg/mL) extract was added to the BHI liquid medium containing HA/TCP, bacteria were inoculated at 1 × 10^8^ CFU/mL, incubated at 37 °C for 24 h, and centrifuged at 1000 rpm for 10 min. The supernatant was extracted for analysis. Then, 100 μL of supernatant was carefully dispensed into a 96-well plate, and the free calcium level was measured in accordance with the protocol of the calcium assay kit (QuantiChrom™ Calcium Assay Kit [DICA-500], BioAssay system, Hayward, CA, USA). The absorbance was measured using an ELISA reader (Molecular Devices Co, San Jose, CA, USA) at 610 nm.

### 4.7. Real-Time Polymerase Chain Reaction (PCR) Analysis of S. mutans Virulence Genes

To evaluate the effect of *C. tricuspidata* (15, 30, 45, and 60 μg/mL) on gene expression of *S. mutans*, real-time PCR analysis was conducted. Total RNA was isolated from *S. mutans*, treated with different concentrations of the extract, and cDNA was synthesized. The StepOnePlus Real-Time PCR System with QPCR SYBR Green Mixes (Applied Biosystem, Foster City, CA, USA) was used for amplification. The expression level of 16S rRNA was used as the control. Primer pairs were reported in a previous study [41] (Table 3). The biological roles of each gene are in Table 4 [5].

### 4.8. UPLC-Q-TOF-MS Analysis

The aqueous extract of the MSJZD sample was analyzed on a Waters ACQUITY UPLC I-Class PLUS system (Waters Corporation, Milford, MA, USA), equipped with a Waters UPLC BEH C18 column (100 mm × 2.1 mm, 1.7 µm particle size), at a column temperature of 40 °C. The mobile phase consisted of acetonitrile (A) and water (B), each containing 0.1% formic acid. The elution procedure was as follows: 99–99% B at 0–1 min; 99–50% B at 1–15 min; 50–40% B at 15–17 min; 40–1% B at 17–18 min; 1% B at 18–21 min. The flow rate was 0.3 mL/min, and the injection volume was 2 μL.

The mass spectrometric data were collected using a time-of-flight analyzer with TurboIonSpray (AB Sciex, Singapore) ion source in negative ion mode. The specific conditions were as follows: nebulizing gas (N2): 55 psi; drying gas (N2): 45 psi; curtain gas: 35 psi; source temperature: 600 °C; ions apart voltage floating: 5500 V/−4500 V; TOF MS scan *m*/*z* range: 50–1500 Da; TOF-MS/MS scan m/z range: 25–1000 Da; TOF MS scan accumulation time: 0.25 s/spectra; product ion scan accumulation time: 0.035 s/spectra. Secondary mass spectrometry was obtained by information-dependent acquisition and high sensitivity mode. Declustering potential was ±60 V; collision energy was 35 ± 15 eV. IDA setup was as follows: Exclude isotopes within 4 Da; candidate ions to monitor per cycle was 12. The data were processed using SCIEX OS software (ver. 3.0) with multiple confidence criteria, including quality accuracy, retention time, isotopes, and matching use of compound libraries. In the current study, the TCM MS/MS Library in the SCIEX OS software was employed to identify the major constituents in MSJZD according to the first-order accurate mass number, isotope distribution ratio, and MS/MS of the constituents.

### 4.9. Statistical Analysis

The experiment was repeated three times, and the results were expressed as the mean and standard deviation (SD) using the statistical program SPSS (ver. 12.0). After analysis of ANOVA, a post hoc test was performed using Tukey HSD. *p* ≤ 0.05 was considered statistically significant.

## 5. Conclusions

This study demonstrated the anti-cariogenic effects of *C. tricuspidata* ethanol extracts against *S. mutans*. Additionally, some effects on gene expression related to biofilm formation of *S. mutans* were observed. The components of the *C. tricuspidata* root ethanol extract included phenylpropanoids, flavonoids, and xanthan; these compounds may account for the anti-cariogenic effect of *S. mutans*. These results suggest that *C. tricuspidata* may inhibit the cariogenic effects associated with the expression of caries-related genes in *S. mutans*.

## Figures and Tables

**Figure 1 molecules-30-01755-f001:**
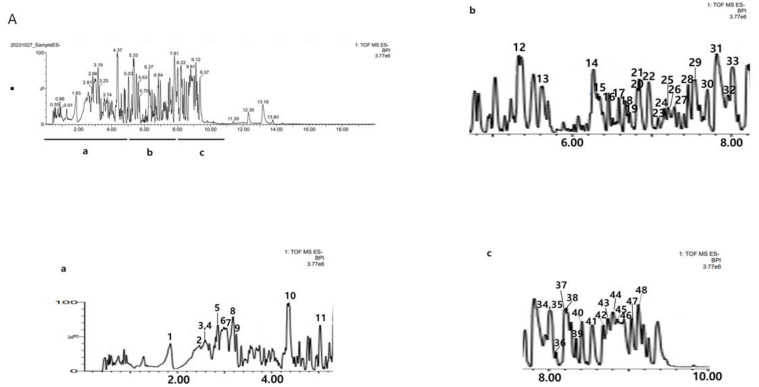
The base peak chromatograms (BPC) of *C. tricuspidata* monitored by UPLC-Q-TOF/MS in negative ion mode. The numbers in (**A**) are RT (min). (**a**) The BPC of RT from 0 to 5 min. (**b**) The BPC of RT from 5 to 8 min. (**c**) The BPC of RT from 8 to 10 min. The numbers in (**a**–**c**) are the peak No. in Table 1.

**Figure 2 molecules-30-01755-f002:**
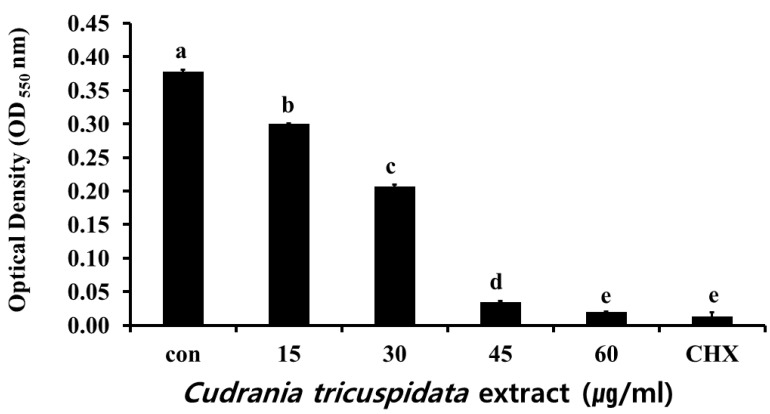
Inhibitory effect of *C. tricuspidata* extract on the growth of *S. mutans*. A degree of inhibition was observed at concentrations of 15, 30, 45, and 60 μg/mL of *C. tricuspidata* extract; 0.05% CHX was used as a positive control. Significance differences among the groups are indicated by different letters (*p* ≤ 0.05).

**Figure 3 molecules-30-01755-f003:**
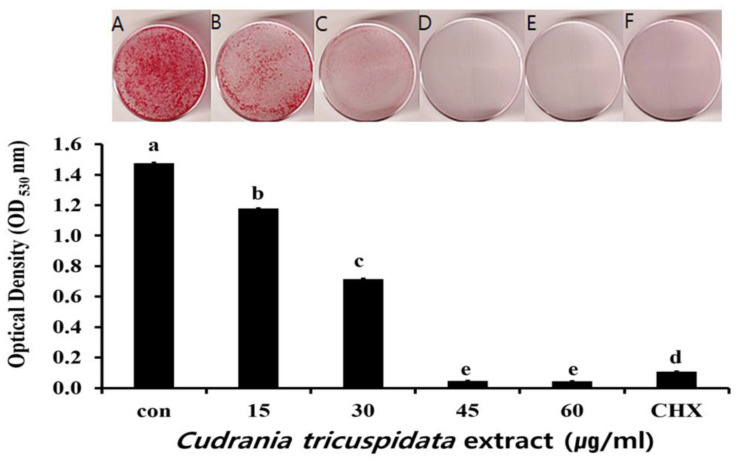
Inhibitory effect of *C. tricuspidata* extract on biofilm formation in *S. mutans*. Safranin staining of *S. mutans* biofilms after treatment with *C. tricuspidata* extract (**A**) control, (**B**) 15 μg/mL, (**C**) 30 μg/mL, (**D**) 45 μg/mL, (**E**) 60 μg/mL, and (**F**) positive control (0.05% CHX). Significance differences among the groups are indicated by different letters (*p* ≤ 0.05).

**Figure 4 molecules-30-01755-f004:**
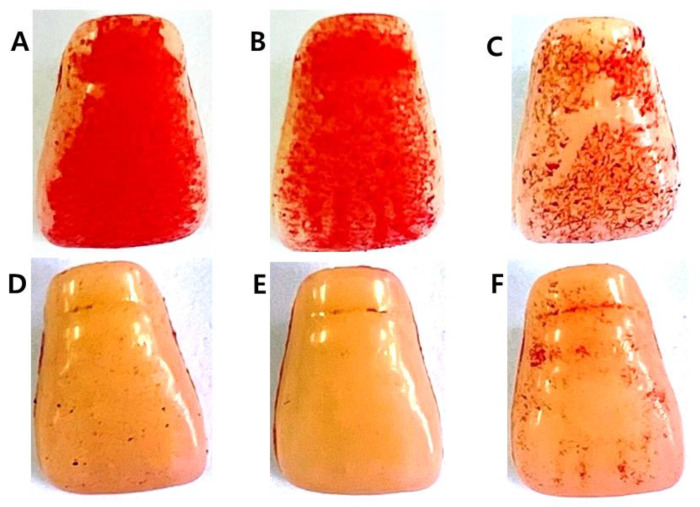
Inhibitory effect of *C. tricuspidata* extract on *S. mutans* biofilm formation on the surface of a resin-based tooth. (**A**) Control, (**B**) 15 μg/mL, (**C**) 30 μg/mL, (**D**) 45 μg/mL, (**E**) 60 μg/mL, and (**F**) positive control (0.05% CHX).

**Figure 5 molecules-30-01755-f005:**
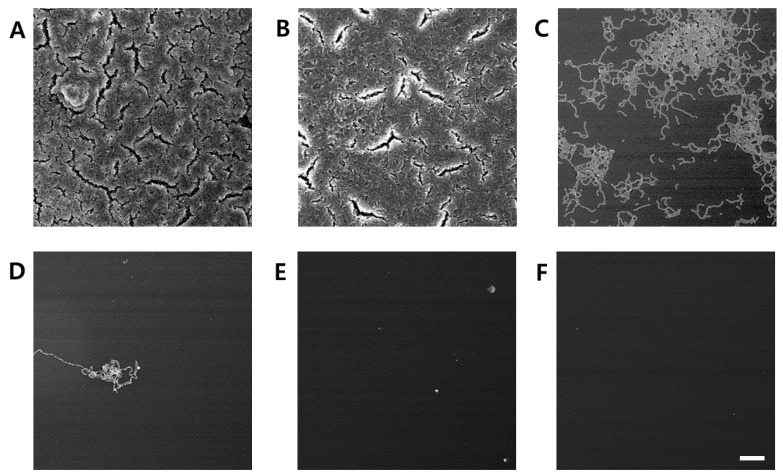
Inhibitory effect of *C. tricuspidata* extract on *S. mutans* biofilm formation observed using scanning electron microscopy. (**A**) Control, (**B**) 15 μg/mL, (**C**) 30 μg/mL, (**D**) 45 μg/mL, (**E**) 60 μg/mL, and (**F**) positive control (0.05% CHX) bar = 100 μm.

**Figure 6 molecules-30-01755-f006:**
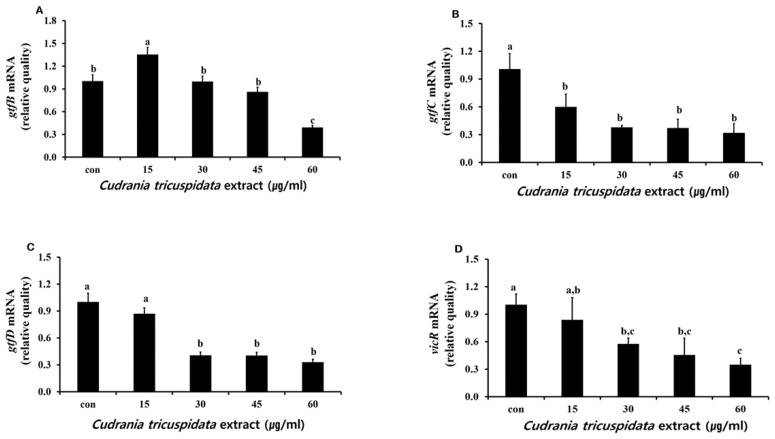
Real-time PCR analysis of multiple genes encoding biofilm formation-associated virulence factors. After culturing *S. mutans* with various concentrations of *C. tricuspidata* extract, real-time PCR analysis of *gtfB* (**A**), *gtfC* (**B**), *gtfD* (**C**), and *vicR* (**D**) was performed. Each value is expressed as mean ± standard deviation. Significance differences among the groups are indicated by different letters (*p* ≤ 0.05).

**Figure 7 molecules-30-01755-f007:**
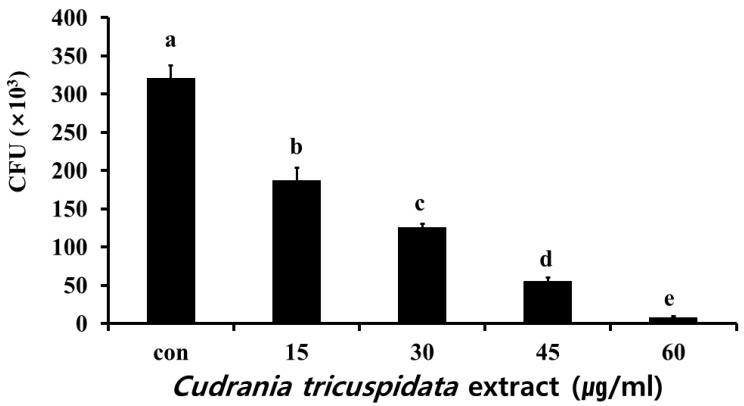
Inhibitory effect of *C. tricuspidata* extract on the adherence of *S. mutans*. Colony forming units (CFU) of *S. mutans* observed after treatment of 30 mg of S-HA beads by various concentrations of *C. tricuspidata* extract. Significance differences among the groups are indicated by different letters (*p* ≤ 0.05).

**Figure 8 molecules-30-01755-f008:**
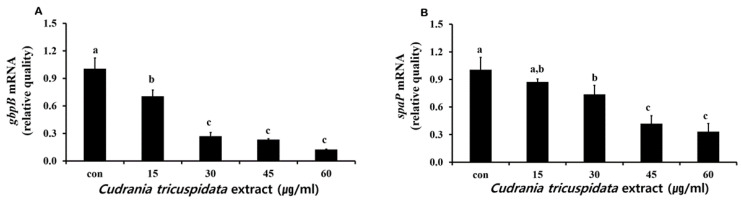
Real-time PCR analysis of the genes (*gbpB* and *spaP*) encoding adherence virulence factors. Real-time PCR analysis of *gbpB* (**A**) and *spaP* (**B**) was performed by culturing *S. mutans* with various concentrations of *C. tricuspidata* extract. Each value is expressed as mean ± standard deviation. Significance differences among the groups are indicated by different letters (*p* ≤ 0.05).

**Figure 9 molecules-30-01755-f009:**
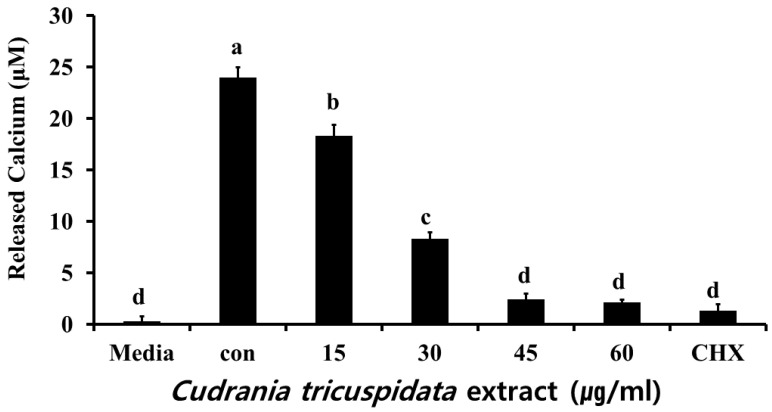
Effect of *C. tricuspidata* extract on calcium release by *S. mutans*. The degree of inhibition of calcium release was observed in the presence of *C. tricuspidata* extract at concentrations of 15, 30, 45, and 60 µg/mL, and CHX (0.05%) was used as a positive control. Significance differences among the groups are indicated by different letters (*p* ≤ 0.05).

**Figure 10 molecules-30-01755-f010:**
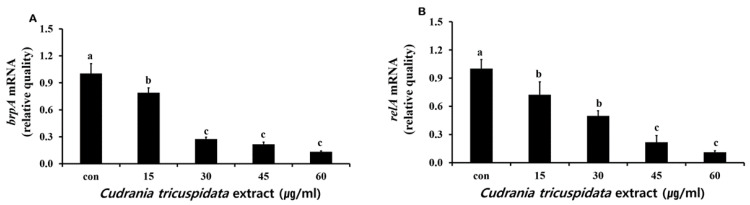
Real-time PCR analysis of *brpA* (**A**) and *relA* (**B**) genes encoding acid resistance-associated virulence factors in *S. mutans* cultured and treated with various concentrations of *C. tricuspidata* extract. Each value is expressed as mean ± standard deviation. Significance differences among the groups are indicated by different letters (*p* ≤ 0.05).

**Table 1 molecules-30-01755-t001:** Compounds identified in *Cudrania tricuspidata* by UPLC-Q-TOF-MS negative ion mode.

Peak No.	RT (min)	Tentative Identification	Mass [M − H]^−^ or [M − HCOO]^−^	Neutral Mass (*m*/*z*)	Molecular Formula	Mass Error (ppm)
			Observed (*m/z*)			
1	1.85	Koaburaside	331.1038	332.1107	C_14_H_20_O_9_	1.1
2	2.46	Asperuloside (isomer I)	459.1157	414.1162	C_18_H_22_O_11_	2.8
3	2.59	Cichoriin	339.0731	340.0794	C_15_H_16_O_9_	2.9
4	2.59	2-Hydroxy-1,4-Naphthoquinone	219.0304	174.0317	C_10_H_6_O_3_	2.4
5	2.81	Asperuloside (isomer II)	459.1167	414.1162	C_18_H_22_O_11_	4.9
6	3.02	Cistanoside B	859.2918	814.2895	C_37_H_50_O_20_	4.7
7	3.07	Viscumneoside V	727.2123	728.2164	C_32_H_40_O_19_	4.4
8	3.18	Jionoside B1	859.2917	814.2895	C_37_H_50_O_20_	4.6
9	3.26	Tenuifoliside A	727.2124	682.2109	C_31_H_38_O_17_	4.5
10	4.36	Khellin	259.0625	260.0685	C_14_H_12_O_5_	5
11	5.03	Gericudranin E (isomer I)	393.0989	394.1053	C_22_H_18_O_7_	2.2
12	5.37	Gericudranin E (isomer II)	393.0996	394.1053	C_22_H_18_O_7_	4.1
13	5.61	Tianshic acid	329.2342	330.2406	C_18_H_34_O_5_	2.4
14	6.27	Cudratricusxanthone L (isomer I)	327.0882	328.0947	C_18_H_16_O_6_	2.3
15	6.35	Licoricone	427.1401	382.1416	C_22_H_22_O_6_	0.7
16	6.46	Mortatarin B (isomer I)	455.1715	456.1784	C_25_H_28_O_8_	0.8
17	6.59	Mortatarin B (isomer II)	455.1711	456.1784	C_25_H_28_O_8_	0
18	6.7	Mortatarin B (isomer III)	455.1711	456.1784	C_25_H_28_O_8_	−0.2
19	6.73	Cudratricusxanthone L (isomer II)	327.0872	328.0947	C_18_H_16_O_6_	−0.6
20	6.82	Cudratricusxanthone L (isomer III)	327.0877	328.0947	C_18_H_16_O_6_	0.8
21	6.85	Cudratricusxanthone M	413.1603	414.1679	C_23_H_26_O_7_	−0.7
22	6.97	Cudratricusxanthone L (isomer IV)	327.0877	328.0947	C_18_H_16_O_6_	1
23	7.08	Mortatarin C (isomer I)	439.1757	440.1835	C_25_H_28_O_7_	−1.1
24	7.16	Mortatarin C (isomer II)	439.1763	440.1835	C_25_H_28_O_7_	0.1
25	7.21	Cudratricusxanthone L (isomer V)	327.0872	328.0947	C_18_H_16_O_6_	−0.7
26	7.24	Mortatarin C (isomer III)	439.1756	440.1835	C_25_H_28_O_7_	−1.3
27	7.34	Cudratricusxanthone L (isomer VI)	327.0872	328.0947	C_18_H_16_O_6_	−0.6
28	7.46	Kuwanon E (isomer I)	423.1815	424.1886	C_25_H_28_O_6_	0.5
29	7.54	5,7,3′,4′-Tetramethoxyflavone	341.1040	342.1103	C_19_H_18_O_6_	2.8
30	7.7	Isoarundinin II (isomer I)	395.1495	350.1518	C_22_H_22_O_4_	−1.2
31	7.82	Isoarundinin II (isomer II)	395.1505	350.1518	C_22_H_22_O_4_	1.2
32	7.88	Cudraflavenone A (isomer I)	421.1657	422.1729	C_25_H_26_O_6_	0.2
33	7.97	Isoarundinin II (isomer III)	395.1500	350.1518	C_22_H_22_O_4_	0
34	8.01	Cudraflavenone A (isomer II)	421.1655	422.1729	C_25_H_26_O_6_	−0.4
35	8.02	Kuwanon E (isomer II)	423.1823	424.1886	C_25_H_28_O_6_	2.4
36	8.09	Bavacoumestan A	351.0873	352.0947	C_20_H_16_O_6_	−0.4
37	8.22	Isoarundinin II (isomer IV)	395.1507	350.1518	C_22_H_22_O_4_	1.6
38	8.24	Cudraflavenone A (isomer III)	421.1662	422.1729	C_25_H_26_O_6_	1.2
39	8.35	Cudraxanthone B (isomer I)	393.1343	394.1416	C_23_H_22_O_6_	−0.1
40	8.42	Kuwanon A	419.1504	420.1573	C_25_H_24_O_6_	1
41	8.55	Isoarundinin II (isomer V)	395.1500	350.1518	C_22_H_22_O_4_	0
42	8.68	Isoarundinin II (isomer VI)	395.1501	350.1518	C_22_H_22_O_4_	0.2
43	8.74	Isoarundinin II (isomer VII)	395.1516	350.1518	C_22_H_22_O_4_	4.1
44	8.81	Isoarundinin II (isomer IX)	395.1520	350.1518	C_22_H_22_O_4_	5
45	8.82	Cudraxanthone B (isomer II)	393.1347	394.1416	C_23_H_22_O_6_	0.9
46	8.94	Isoarundinin II	395.1510	350.1518	C_22_H_22_O_4_	2.6
47	9.04	Cudraxanthone B (isomer III)	393.1348	394.1416	C_23_H_22_O_6_	1
48	9.19	Isoarundinin II	395.1503	350.1518	C_22_H_22_O_4_	0.8

**Table 2 molecules-30-01755-t002:** pH changes in *S. mutans* cultures incubated with different concentrations of *C. tricuspidata* extract.

Con. (μg/mL)	pH (Before Incubation)	pH (After Incubation)
Control	7.33 ± 0.04 ^a^	5.39 ± 0.10 ^a^
15	7.30 ± 0.02 ^a^	5.32 ± 0.12 ^a^
30	7.32 ± 0.02 ^a^	6.26 ± 0.04 ^b^
45	7.34 ± 0.06 ^a^	7.13 ± 0.04 ^c^
60	7.31 ± 0.04 ^a^	7.17 ± 0.07 ^c^
CHX (0.05%)	7.34 ± 0.07 ^a^	7.15 ± 0.05 ^c^

Data (pH) are represented as the mean ± SD. Significance differences among the groups are indicated by different letters (*p* ≤ 0.05).

**Table 3 molecules-30-01755-t003:** Oligonucleotide primers used in this study.

Genes *	Primer Sequences (5’-3’)	
16S rRNA	Forward	CCTACGGGAGGCAGCAGTAG
	Reverse	CAACAGAGCTTTACGATCCGAAA
*gbpB*	Forward	ATGGCGGTTATGGACACGTT
	Reverse	TTTGGCCACCTTGAACACCT
*spaP*	Forward	GACTTTGGTAATGGTTATGCATCAA
	Reverse	TTTGTATCAGCCGGATCAAGTG
*gtfB*	Forward	AGCAATGCAGCCAATCTACAAAT
	Reverse	ACGAACTTTGCCGTTATTGTCA
*gtfC*	Forward	GGTTTAACGTCAAAATTAGCTGTATTAGC
	Reverse	CTCAACCAACCGCCACTGTT
*gtfD*	Forward	ACAGCAGACAGCAGCCAAGA
	Reverse	ACTGGGTTTGCTGCGTTTG
*vicR*	Forward	TGACACGATTACAGCCTTTGATG
	Reverse	CGTCTAGTTCTGGTAACATTAAGTCCAATA
*relA*	Forward	ACAAAAAGGGTATCGTCCGTACAT
	Reverse	AATCACGCTTGGTATTGCTAATTG
*brpA*	Forward	GGAGGAGCTGCATCAGGATTC
	Reverse	AACTCCAGCACATCCAGCAAG

* Based on the NCBI *S. mutans* genome database.

**Table 4 molecules-30-01755-t004:** The functions of virulence genes in Streptococcus mutans *.

Function	Gene	Biological Roles
Adhesion	*gbpB*	Encode glucan binding proteins (GBP) A GBP in the cell membrane of *S. mutans* plays an important role in adherence of *S. mutans* to glucan molecules.
	*spaP*	Encodes cell surface antigen (SpaP) Adheres to salivary agglutinin glycoprotein (SAG) and proline-rich protein of the acquired pellicle on the tooth surface as a kind of surface fibrillar adhesin.
Formation of extracellular polysaccharide in biofilm	*gtfB*, *gtfC*, *gtfD*	Encode glycosyltransferase (GTF) B, GTF C, and GTF D. Synthesize glucan by polymerizing glucose.
Regulation	*vicR*	Encodes putative histidine kinase. Regulates expression of *gtfB*, *gtfC*, and *gtfD.*
Sugar uptake and metabolism	*relA*	Contributes to the regulation of glucose phosphotransferase system (PTS), the glucose uptake system of *S. mutans*
Acid tolerance	*brpA*	Contributes to acid tolerance

* ref [5] Yong-Ouk You. Int J Oral Biol 44:31–36, 2019.

## Data Availability

The data that support the findings of this study are available from the corresponding author upon reasonable request.

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
