# Peer review of "Effect of Cudrania tricuspidata on Cariogenic Properties and Caries-Related Gene Expression in Streptococcus mutans"

_molecules, 2025, doi:10.3390/molecules30081755_

Round 1
Reviewer 1 Report
Comments and Suggestions for Authors
I have gone through the manuscript ID: molecules-3557017 entitled: “Effect of Cudrania tricuspidata on cariogenic propertiesand caries-related gene expresson in Streptococcus mutans”. Even though the subject of the research is interesting and important, there some major stuff in the manuscript that need to be address. I have listed my comments below.
- Please change in the title „expresson“ to „expression“.
- Throughout the whole manuscript put bacterial strains and genes in italic.
- English language needs to be improved in the whole manuscript. Some sentences are a bit confusing due to the insufficiently good English.
- Lines 78-80: This sentence does not go with the rest of the introduction. Here you gave information about another plant.
- It would be good to put at the beginning of the results the chemical analysis of extract, and then to discuss about compounds present in extract in terms of observed activities.
- Please indicate which part of the plant is used for extraction.
- Overall, methodology is a bit confusing, it needs to be more clearly and more detailed written. Please indicate what concentrations were used in every method.
- It is really hard to follow results now. It would be good to put the all results of real time PCR as a separate subsection.
- Lines 211-219: This part should be used when discussing your results, it should not be at the begging of discussion.
- The whole discussion should be redone. The chemical analysis should be discussed first, so that you can relate the observed effects to the compounds found in the extract. In general, the discussion is weak, there is more repetition of the results, and it is necessary to explain the obtained results and cite more literature.
Comments on the Quality of English Language
English is a bit confusing. The sentences in certain places in manuscript are not clear and are very hard to follow. The whole manuscript needs to go through English language revision.
Author Response
Title: Effect of Cudrania tricuspidata on cariogenic properties and caries-related gene expression in Streptococcus mutans
Summary
Thank you very much for taking the time to review this manuscript. Please find the detailed responses below and the corrections highlighted (red color) in the re-submitted files.
Reviewer 1
Comments 1: Please change in the title „expresson“ to „expression“.
Response 1: The title has been corrected as suggested by the reviewer.(line 2)
Comments 2: Throughout the whole manuscript put bacterial strains and genes in italic.
Response 2: The bacterial strains and genes were put in italics as suggested by the reviewer.
(line 32-57)
Comments 3: English language needs to be improved in the whole manuscript. Some sentences are a bit confusing due to the insufficiently good English.
Response 3: Based on the reviewer's suggestion, I requested 'Editage’ (English editing services) to revise the entire manuscript. Editing Certificate is attached.
Comments 4: Lines 78-80: This sentence does not go with the rest of the introduction. Here you gave information about another plant.
Response 4: The euchrestaflavanone ingredient part has been deleted and revised as suggested by the reviewer.(line 71-72)
Comments 5: It would be good to put at the beginning of the results the chemical analysis of extract, and then to discuss about compounds present in extract in terms of observed activities.
Response 5: As suggested by the reviewer, the results of the chemical analysis were placed at the beginning and discussed.(line 79-92, 196-207)
Comments 6: Please indicate which part of the plant is used for extraction.
Response 6: The part of the plant (Root) has been added as suggested by the reviewer.(line 79,263)
Comments 7: Overall, methodology is a bit confusing, it needs to be more clearly and more detailed written. Please indicate what concentrations were used in every method.
Response 7: The methodology has been revised as suggested by the reviewer.(line 273-330) The concentrations used have been added.
Comments 8: It is really hard to follow results now. It would be good to put the all results of real time PCR as a separate subsection.
Response 8: The gbpB and spaP genes are involved in the mechanisms of bacterial adhesion. The mechanisms by which C. tricuspidata inhibits the adhesion of S. mutans may involve the suppression of gbpB and spaP gene expression. Therefore, the data on gbpB and spaP gene expression were presented after the bacterial adhesion results.
The gtfB, gtfC, and gtfD genes are associated with the formation of extracellular polysaccharides in S. mutans biofilms. Therefore, gtfB, gtfC, and gtfD are crucial for biofilm formation. The mechanisms by which C. tricuspidata inhibits the biofilm formation of S. mutans may involve the suppression of gtfB, gtfC, and gtfD gene expression. Consequently, the expression data of gtfB, gtfC, and gtfD genes were presented after the biofilm formation assay.
The relA and brpA genes are involved in sugar metabolism, acid production, and acid tolerance in S. mutans. Hence, the relA and brpA gene expression data were arranged after the acid production results.
Comments 9: Lines 211-219: This part should be used when discussing your results, it should not be at the begging of discussion.
Response 9: Lines 211-219 have been deleted as suggested by the reviewer. The discussion of that part is moved to lines 251-256.
Comments 10: The whole discussion should be redone. The chemical analysis should be discussed first, so that you can relate the observed effects to the compounds found in the extract. In general, the discussion is weak, there is more repetition of the results, and it is necessary to explain the obtained results and cite more literature.
Response 10: The whole discussion has been revised as suggested by the reviewer.
(line 191-262)

Reviewer 2 Report
Comments and Suggestions for Authors
the manuscript describes the effect of the extract from C. tricuspidata considering several aspects involved in cariogenic development. The results are very interesting and promising.
The manuscript provides a clear data presentation and discussion, therefore it can be accepted after a minor revision.
I suggest to consider the following points:
A repetition of the same section seems to be present; check sections 4.2 e 4.3
Line 106 _figure 2-3-4 or 2-4. In my opinion 2.3.4 can be misleading
Line 260: please add a reference regarding propolis
Author Response
Reviewer 2
Comments 1: A repetition of the same section seems to be present; check sections 4.2 e 4.3
Response 1: Sections 4.2 and 4.3 have been written as one section as suggested by the reviewer.(line 271-277)
Comments 2: Line 106 _figure 2-3-4 or 2-4. In my opinion 2.3.4 can be misleading
Response 2: Revised as suggested by the reviewer.(line 111)
Comments 3: Line 260: please add a reference regarding propolis
Response 3: References have been added as suggested by the reviewer.(line 251,254)

Reviewer 3 Report
Comments and Suggestions for Authors
Major points
- an exhaustive English and text formatting revision is required for the whole manuscript
- bacterial names should utilise the correct nomenclature in italics – e.g., Streptococcus mutans
- genes should also utilise the correct nomenclature in italics – e.g., “Genes related to adherence include spaP” - spaP
- introduction – add a few ideas regarding the primary and secondary bacterial colonisers involved in the cariogenic process
Methods
- line 329 – add a brief description of the protocol
- table 3 – add a brief description of the biological roles of each protein coded by the examined genes
- line 378 – “At the p = 0.05 level, the average value of the experimental group and the control group was verified by an independent sample t-test.” – this is unclear; was the statistical significance set at p ≤ 0.05? furthermore, were all groups compared with the control group through independent t tests? Why was one-way ANOVA not chosen, given the fact that there are multiple groups?
Results
- section 2.2 the methods section only describes the biofilm evaluation on artificial teeth; also add the methodology regarding the evaluation on the dish
- section 2.2 it is somewhat unclear how the results were obtained and compared to each other in order to apply statistical tests?
- line 105 – “S. mutans biofilm formation was significantly high at concentrations ≥45 μg/mL (p <0.05) (Figure 2.3.4)” – which Figure is Figure 2.3.4? If the reference is to Figure 2,3,4 change the punctuation
- line 108 – “The expression of gtfB and gtfC was inhibited significantly by the extract in a concentration-dependent manner at ≥45 μg/mL and ≥15 μg/mL, respectively, while the extract concentrations ≥30 μg/mL significantly inhibited the expression of vicR (p <0.05) (Figure 5).” – in order to affirm that the extract acted in a dose dependent manner each dose should be compared to the previous dose and not the control group; furthermore, the results presented in the text seem to not match the graphs – e.g., gtfB seems significantly decreased by all doses; this whole results section requires rephrasing
- line 139 – “The expression of gbpB and spaP was significantly inhibited at concentrations ≥15 μg/mL and ≥45 μg/mL, respectively” – same as above; furthermore, there is no mention in the text of spaP
- Figure 10 could be uploaded in a higher resolution to improve readability
- references are not in the correct format
Minor points
- line 68 – not all readers may be familiar with Donguibogam (briefly explain that it relates to Korean traditional medicine)
- line 79 – the full scientific name of the bacterium should be used in its first occurrence within the text (S. iniae)
- figure 5 – gtfB and not ghpB
- line 235 – “and causes dental caries” – glucan itself does not cause caries; biofilm deposited by S. mutans will aid in the adherence of further bacteria that drive the cariogenic process
Comments on the Quality of English Languagean exhaustive English and text formatting revision is required for the whole manuscript
Author Response
Reviewer 3
Comments 1: an exhaustive English and text formatting revision is required for the whole manuscript
Response 1: Based on the reviewer's suggestion, I requested 'Editage’ (English editing services) to revise the entire manuscript. Editing Certificate is attached.
Comments 2: bacterial names should utilise the correct nomenclature in italics – e.g.,Streptococcus mutans
Response 2: The bacterial strains was put in italics as suggested by the reviewer.
(line 32-57)
Comments 3: genes should also utilise the correct nomenclature in italics – e.g., “Genes related to adherence include spaP” -spaP
Response 3: genes were put in italics as suggested by the reviewer.(line 32-57)
Comments 4: introduction – add a few ideas regarding the primary and secondary bacterial colonisers involved in the cariogenic process
Response 4: Added to the introduction based on the reviewer's suggestion.(line 32-41)
Methods
Comments 5: line 329 – add a brief description of the protocol
Response 5: A brief description was added based on the reviewer's suggestion.(line 303-306)
Comments 6: table 3 – add a brief description of the biological roles of each protein coded by the examined genes
Response 6: As suggested by the reviewer, the biological roles have been described in Table 4.(line 338-340)
Comments 7: line 378 – “At the p = 0.05 level, the average value of the experimental group and the control group was verified by an independent sample t-test.” – this is unclear; was the statistical significance set at p ≤ 0.05? furthermore, were all groups compared with the control group through independent t tests? Why was one-way ANOVA not chosen, given the fact that there are multiple groups?
Response 7: All data were reanalyzed by ANOVA at the reviewer’s suggestion, and post hoc tests were performed with Tukey HSD. The statistical significance was set at p ≤ 0.05.(line 367-370)
Results
Comments 8: section 2.2 the methods section only describes the biofilm evaluation on artificial teeth; also add the methodology regarding the evaluation on the dish
Response 8: At the reviewer's suggestion, the dish evaluation content has been added to section 2.2. The protocol for artificial teeth and dishes is the same, so the dish has been supplemented.(line 284-290)
Comments 9: section 2.2 it is somewhat unclear how the results were obtained and compared to each other in order to apply statistical tests?
Response 9: The statistical test data in section 2.2 supplemented the method of measuring the dye solution in the dishes. The sentence has been revised as suggested by the reviewer. (line 285-287)
Comments 10: line 105 – “S. mutans biofilm formation was significantly high at concentrations ≥45 μg/mL (p <0.05) (Figure 2.3.4)” – which Figure is Figure 2.3.4? If the reference is to Figure 2,3,4 change the punctuation
Response 10: Revised as suggested by the reviewer.(line 112)
Comments 11: line 108 – “The expression of gtfB and gtfC was inhibited significantly by the extract in a concentration-dependent manner at ≥45 μg/mL and ≥15 μg/mL, respectively, while the extract concentrations ≥30 μg/mL significantly inhibited the expression of vicR (p <0.05) (Figure 5).” – in order to affirm that the extract acted in a dose dependent manner each dose should be compared to the previous dose and not the control group; furthermore, the results presented in the text seem to not match the graphs – e.g., gtfB seems significantly decreased by all doses; this whole results section requires rephrasing
Response 11: The figure was inserted incorrectly and has been corrected to the correct figure. And the results have been discussed correctly.(line 113-117)
Comments 12: line 139 – “The expression of gbpB and spaP was significantly inhibited at concentrations ≥15 μg/mL and ≥45 μg/mL, respectively” – same as above; furthermore, there is no mention in the text of spaP
Response 12: Revised as suggested by the reviewer. The content regarding the gbpB and spaP genes is supplemented in the Discussion.(line 226-243)
Comments 13: Figure 10 could be uploaded in a higher resolution to improve readability
Response 13: I will upload the original file of Figure 10.
Comments 14: references are not in the correct format
Response 14: The reference format was revised as suggested by the reviewer.
Minor points
Comments 15: line 68 – not all readers may be familiar with Donguibogam (briefly explain that it relates to Korean traditional medicine)
Response 15: Since Donguibogam is a Korean traditional medicine book, it has been revised to Korean traditional medicine.(line 65)
Comments 16: line 79 – the full scientific name of the bacterium should be used in its first occurrence within the text (S. iniae)
Response 16: The content related to S. iniae has been deleted by a brief summary.(line 72-73)
Comments 17: figure 5 – gtfB and not ghpB
Response 17: Inserted as an accurate figure.(line 133)
Comments 18:
line 235 – “and causes dental caries” – glucan itself does not cause caries; biofilm deposited by S. mutans will aid in the adherence of further bacteria that drive the cariogenic processResponse 18: The sentence has been revised as suggested by the reviewer. (line 221-223)

Round 2
Reviewer 1 Report
Comments and Suggestions for Authors
The authors responded to all of my comments. I suggest this paper for publishing now.
Reviewer 3 Report
Comments and Suggestions for Authors
I have no further comments. In my opinion, the manuscript has been sufficiently improved during revisions